# The efficacy of preoperative evolocumab-rosuvastatin combination therapy in patients with ST-elevation myocardial infarction

Hongxu Li[1], Huawei Dong[2], Shilin Bao[2], Xuemei Wang[2], Zuolin Fu🔴[2]*

**1** Shandong First Medical University, Jinan, Shandong, China, **2** Liaocheng Hospital Affiliated to Shandong First Medical University, Liaocheng, Shandong, China

\* fzl04@163.com

## Abstract

### Background

The impact of single-dose preoperative evolocumab combined with rosuvastatin therapy prior to emergency percutaneous coronary intervention (PCI) in patients with acute ST-elevation myocardial infarction (STEMI) remains insufficiently characterized within current guideline-directed medical therapy.

### Methods

In this prospective randomized trial conducted at Liaocheng People's Hospital (2023–2024), 80 STEMI patients undergoing emergency PCI were randomized to: Treatment group: Single subcutaneous evolocumab 140 mg plus oral rosuvastatin 10 mg administered pre-PCI, followed by rosuvastatin 10 mg/day; Control group: Rosuvastatin 10 mg/day alone initiated post-PCI. Primary endpoint was major adverse cardiovascular events (MACEs) at 6 months. Secondary endpoints included angina incidence, low-density lipoprotein cholesterol (LDL-C) levels, interleukins, and ST-segment resolution rate (STR). The trial was registered at the Chinese Clinical Trial Registry (ChiCTR2500099498).

### Results

Primary endpoint (MACEs): 5.0% (treatment group) vs. 12.5% (control group) ($P = 0.228$) at 6-month follow-up. Secondary endpoints: Angina incidence: 7.5% vs 27.5% ($P = 0.037$) at 6-month follow-up; LDL-C reduction: Significant in treatment group at day 1 ($2.97 \pm 0.63$ vs $3.33 \pm 0.78$ mmol/L; $P = 0.029$), day 7 ($1.66 \pm 0.89$ vs $2.25 \pm 0.77$ mmol/L, $P = 0.003$), and month 1 ($P = 0.036$); ST-segment resolution >70%: 60% vs 30% ($P < 0.05$); Inflammatory markers: Lower IL-6 ($P = 0.02$) and IL-17 ($P = 0.01$) in treatment group.

**Data availability statement:** Full data will be deposited in [Figshare, http://Figshare.com], DOI: https://doi.org/10.6084/m9.figshare.29986402.

**Funding:** The author(s) received no specific funding for this work.

**Competing interests:** The authors have declared that no competing interests exist.

## Conclusions

While the evolocumab-rosuvastatin combination did not significantly reduce 6-month MACEs, it demonstrated clinically important benefits including reduced angina frequency, accelerated LDL-C lowering, improved myocardial reperfusion, and attenuated inflammatory response, with a favorable safety profile. These findings support further investigation of intensive lipid-lowering strategies in acute STEMI management.

## 1. Introduction

Acute ST-segment elevation myocardial infarction (STEMI), characterized by high morbidity and mortality, primarily results from rupture of unstable atherosclerotic plaques causing coronary artery occlusion and subsequent myocardial necrosis [1]. Despite significant improvements in prognosis through emergency reperfusion therapy, residual risks including ischemia-reperfusion injury, inflammatory responses, and microcirculatory dysfunction persist as major factors affecting clinical outcomes. Beyond pharmacological therapy, a patient's psychological adaptation, such as low resilience, has been linked to poorer adherence to cardiac rehabilitation and higher levels of anxiety, which may indirectly influence long-term clinical outcomes [2]. Studies have shown that the mortality rate of acute myocardial infarction patients has continued to rise over the past decade in China. [3].

Low-density lipoprotein cholesterol (LDL-C) demonstrates a strong atherogenic effect, with studies establishing a linear relationship between cardiovascular adverse events and LDL-C levels [4]. Current guidelines recommend statins as first-line therapy, with adjunctive cholesterol absorption inhibitors and/or proprotein convertase subtilisin/kexin type 9 (PCSK9) inhibitors when necessary [5].

Evolocumab (PCSK9 inhibitor) and rosuvastatin exhibit complementary mechanisms for LDL-C reduction and possess pleiotropic effects including anti-platelet, anti-atherosclerotic, vascular endothelial protection, anti-inflammatory and plaque-stabilizing properties [6–8]. While long-term combination therapy reduces cardiovascular events [9], the impact of single-dose preoperative administration remains unexplored. This trial investigates the efficacy and safety of single-dose evolocumab-rosuvastatin combination before PCI in STEMI patients. Medium-dose rosuvastatin (10 mg) was selected to balance efficacy and safety in Chinese populations, as higher doses may increase hepatotoxicity risk without proven incremental benefits in acute settings [5].

## 2. Materials and methods

### 2.1 Study design and ethics

A prospective randomized controlled trial was conducted at Liaocheng People's Hospital between 2023–2024. The dates of start and end of the recruitment period for this study are July 23, 2023 and April 22, 2024, respectively. The last follow-up date was October 26, 2024. The study protocol complied with the revised Helsinki Declaration and Good Clinical Practice guidelines, approved by the Institutional Ethics Committee

(Approval No. LCSY-2023009). All participants provided written informed consent. This trial was not pre-registered due to initial failure to fully recognize the importance of early registration and lack of familiarity with the registration process. However, a retrospective registration was conducted at the Chinese Clinical Trials Registry (ChiCTR2500099498) before data was analysed, and all data were ensured complete, transparent, and non-selective reporting. The authors confirm that all ongoing and related trials for this drug/intervention are registered. Reporting follows CONSORT guidelines.

## 2.2 Participants

Eligible patients (n = 80) met the following criteria: 1) Age 18–75 years; 2) Diagnosis of acute STEMI per 2019 Chinese Society of Cardiology guidelines [10]; 3) Successful PCI within 12 hours of symptom onset; 4) No prior PCSK9 inhibitor use and statin-free for 30 days.

Exclusion criteria: 1) Severe heart failure (Killip class III-IV) or hemodynamic instability; 2) Hepatic dysfunction (ALT > 120 U/L) or renal impairment (eGFR < 30 mL/min/1.73m²); 3) Active malignancy or systemic inflammatory disorders; 4) moderate to severe anemia (hemoglobin <90 g/L); 5) Contraindications to study medications; 6) currently participating in other clinical trials.

The participants were randomly divided into a control group and a treatment group. Randomization was performed using the random number table method by an independent statistician. Allocation concealment was maintained via sealed opaque envelopes opened after consent

## 2.3 Interventions

Treatment group: Single dose of evolocumab 140 mg (Amgen) subcutaneously + rosuvastatin 10 mg (IPR Pharmaceuticals) orally administered pre-PCI, followed by rosuvastatin 10 mg/d.

Control group: Rosuvastatin 10 mg/d initiated post-PCI.

All patients received standardized care per 2023 Chinese Lipid Guidelines [5], including antiplatelet therapy (aspirin + clopidogrel/ticagrelor), anticoagulation, and individualized revascularization.

## 2.4 Outcome measures

Primary endpoint: Composite major adverse cardiovascular events (MACEs) at 6 months (cardiac death, recurrent MI, unplanned revascularization, stroke, ischemia-driven rehospitalization).

Secondary endpoints: 1) Angina incidence (Those who have an angina attack but do not require hospitalization) ; 2) LDL-C dynamics (postoperative days 1, 7; months 1, 3, 6); 3) Inflammatory markers (IL-6, IL-17 postoperative days 1); 4) ST-resolution rate (STR at 90 min post-PCI).

Angina was assessed according to CCS classification criteria through scheduled clinic visits and documented using standardized case report forms

Safety endpoints: liver/kidney function, myopathy

## 2.5 Statistical analysis

Sample size was calculated based on the primary endpoint of 6-month MACE incidence, assuming 80% power to detect a 25% absolute reduction between groups ($\alpha = 0.05$, two-sided). requiring 38 per group. We enrolled 40 per group to account for attrition. Data analyzed using SPSS 27.0 with intention-to-treat principle. Continuous variables expressed as mean±SD (normal distribution) or median[IQR] (non-normal), compared by Student's t-test or Mann-Whitney U-test. Categorical variables were presented as frequencies (%) and compared by $\chi^2$ or Fisher's exact test. The cumulative incidence of MACEs was assessed using the Kaplan-Meier survival curve, Statistical significance set at two-tailed $P < 0.05$.

Due to the nature of interventions, participants and clinicians were not blinded. However, outcome assessors and statisticians were masked to group assignment throughout data collection and analysis.

## 3. Results

Participant flow is summarized in Fig 1. Between July 2023 and April 2024, 112 STEMI patients were screened, with 80 meeting inclusion criteria and randomized.

### 3.1 Baseline characteristics

No significant differences existed between groups in age, sex, comorbidities, or laboratory parameters (all P > 0.05), confirming successful randomization (Table 1).

### 3.2 Efficacy outcomes

**3.2.1 LDL-C reduction (Table 2).** Treatment group showed significantly lower LDL-C at day 1 (2.97 ± 0.63 vs. 3.33 ± 0.78 mmol/L; P = 0.029), day 7 (1.66 ± 0.89 vs. 2.25 ± 0.77 mmol/L; P = 0.003), and month 1 (P = 0.036). Differences became non-significant at months 3 and 6.

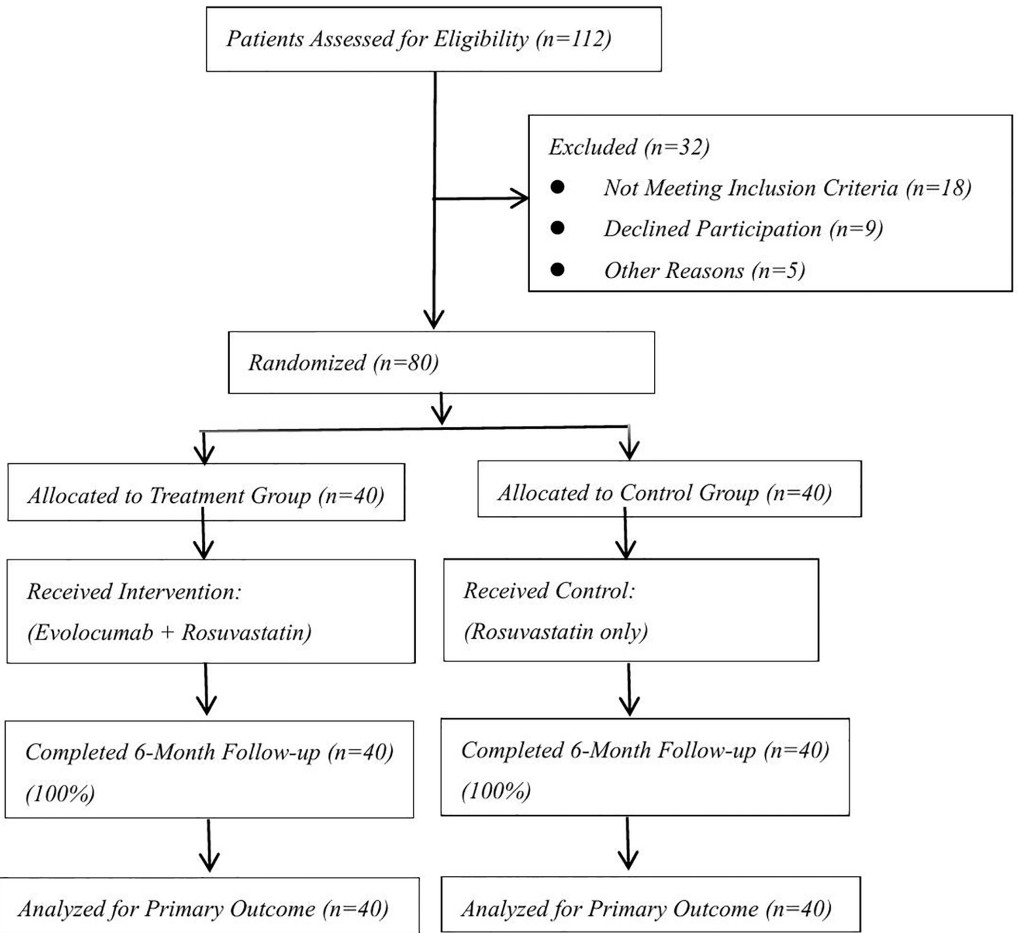

**Fig 1. Completed CONSORT flow diagram Flow of participants through each stage of the randomized trial.** The diagram adheres to the CONSORT 2010 statement. A total of 112 acute ST-elevation myocardial infarction (STEMI) patients were screened between July 2023 and April 2024 at Liaocheng People's Hospital. After exclusions (n = 32), 80 patients were randomized to intervention (n = 40) or control (n = 40). All participants completed the 6-month follow-up and were included in the intention-to-treat (ITT) analysis.

**Table 1. Baseline Characteristics.**

| | | | $n$ (%), $\overline{X} \pm S$ |
|---|---|---|---|
| project | control group(n = 40) | treatment group(n = 40) | P |
| Age (years) | 60.4 ± 11.2 | 59.20 ± 11.0 | 0.623 |
| Male sex, n(%) | 34(85.0) | 37(92.5) | 0.492 |
| Hypertension, n(%) | 20(50) | 20(50) | 1.00 |
| Creatinine (µmol/L) | 63.8 ± 14.4 | 60.3 ± 12.2 | 0.24 |
| Hemoglobin (g/L) | 144.9 ± 13.6 | 147.2 ± 13.1 | 0.46 |
| Alanine aminotransferase(U/L) | 42.5 ± 26.9 | 53.3 ± 50.6 | 0.39 |

**Table 2. Changes in LDL-C levels after treatment.**

| | | | | | $n$(%), $\overline{X} \pm S$ |
|---|---|---|---|---|---|
| | control group | treatment group | MD (95% CI) | t/χ² | P |
| Postoperative day 1 (mmol/ L) | 3.33 ± 0.78 | 2.97 ± 0.63 | 0.35(0.04 ~ 0.70) | 2.23 | 0.029 |
| Postoperative day 7 (mmol/(L) | 2.25 ± 0.77 | 1.66 ± 0.89 | 0.58(0.21 ~ 0.96) | 3.11 | 0.003 |
| Postoperative month 1 (mmol/(L) | 1.99 ± 0.63 | 1.69 ± 0.63 | 0.30(0.02 ~ 0.59) | 2.13 | 0.036 |
| Postoperative month 3 (mmol/(L) | 1.72 ± 0.67 | 1.55 ± 0.63 | 0.17(−0.12 ~ 0.47) | 1.18 | 0.242 |
| Postoperative month 6 (mmol/(L) | 1.63 ± 0.56 | 1.67 ± 0.67 | −0.03(−0.32 ~ 0.25) | −0.25 | 0.801 |

**3.2.2 ST-segment resolution.** Treatment group had higher STR (75.25 ± 18.98% vs. 63.58 ± 21.29%; P = 0.01) and more patients achieving STR > 70% (60% vs. 30%; P < 0.05).

**3.2.3 Inflammatory markers.** Treatment group exhibited significantly lower IL-6 (4.62[2.78–9.95] vs. 9.97[4.19–18.97] pg/mL; P = 0.02) and IL-17 (1.73[0.68–2.29] vs. 2.55[1.39–3.75] pg/mL; P = 0.01) postoperative days 1.

**3.2.4 Clinical endpoints (Table 3 and 4).** All reported angina episodes were physician-adjudicated and required typical symptoms of ischemia with or without objective evidence of ischemia (ECG changes or imaging evidence).

Angina incidence: 3 (7.5%) vs. 11 (27.5%) (P=0.037).

MACEs: 2 (5.0%) vs. 5 (12.5%) (P=0.228).

**Table 3. Angina incidence.**

| | | | n(%) |
|---|---|---|---|
| | Whether angina pectoris occurs | | P |
| | yes | no | |
| control group | 11 | 29 | 0.037 |
| treatment group | 3 | 37 | |

**Table 4. Primary outcome analysis.**

| | | | n(%) |
|---|---|---|---|
| | major adverse cardiovascular events | | P |
| | yes | no | |
| control group | 5 | 35 | 0.228 |
| treatment group | 2 | 38 | |

### 3.3 Safety outcomes

No cases of myopathy, significant transaminase elevation ($3 \times$ ULN). Renal function remained stable in both groups ($P > 0.05$).

## 4. Discussion

This study demonstrates that preoperative administration of evolocumab combined with medium-dose rosuvastatin in STEMI patients: 1) Achieved rapid LDL-C reduction within 24h (27% lower than control); 2) Significantly improved myocardial perfusion (STR > 70%: 60% vs. 30%, $P < 0.05$) potentially through anti-inflammatory effects; 3) Reduced angina incidence by 73% (7.5% vs. 27.5%, $P = 0.037$); 4) Exhibited favorable safety with no increased adverse events.

The neutral MACE outcome (5.0% vs. 12.5%, $P = 0.228$) may reflect: 1) Limited statistical power (Type II error risk due to sample size), 2) Short follow-up duration (6 months) insufficient for MACE detection, 3) Early intervention focus where angina reduction precedes hard endpoints.

Our finding of rapid LDL-C reduction is consistent with EPIC-STEMI findings [11] and studies that are critical to achieving lipid lowering as soon as possible in patients with STEMI [12], Rapid LDL-C reduction within 24 hours may attenuate plaque inflammation and microvascular dysfunction, potentially explaining the observed STR improvement. IL-6 reduction supports the anti-inflammatory mechanism of PCSK9 inhibitors [13], potentially explaining improved microvascular perfusion. The 73% angina reduction suggests symptomatic benefits even without MACE improvement, which is clinically relevant for quality of life.

Recent trials with bempedoic acid [14] have demonstrated additional LDL-C lowering options for statin-intolerant patients. While our study focused on PCSK9 inhibition, the potential synergistic effects of combining bempedoic acid with evolocumab warrants investigation, particularly given their complementary mechanisms targeting cholesterol biosynthesis and clearance respectively. Future studies should explore whether triple therapy (statin+bempedoic acid+PCSK9 inhibitor) could provide incremental benefits in acute coronary syndromes.

Furthermore, future longitudinal studies should explore how interventions combining intensive lipid-lowering therapy with psychosocial support, such as those aimed at strengthening coping skills and resilience, could potentially enhance overall patient outcomes and reduce cardiovascular event recurrence.

Limitations: 1) Single-center design limits generalizability, 2) Underpowered for MACE assessment (post-hoc power = 42%), 3) Lack of long-term follow-up beyond 6 months, 4) Open-label design (though endpoint assessors were blinded)

Future directions: Multicenter RCTs with ≥1,000 participants and 2-year follow-up are needed to confirm MACE benefits.

## 5. Conclusions

In STEMI patients undergoing emergency PCI, single preoperative evolocumab-rosuvastatin combination: 1) did not reduce 6-month MACEs, 2) significantly lowered angina incidence (7.5% vs. 27.5%), 3) provided earlier and more pronounced LDL-C reduction, 4) improved ST-segment resolution and suppressed inflammation, 5) demonstrated excellent safety at medium statin dose.

This regimen represents a promising option for rapid lipid control and symptom improvement in acute STEMI.

## Supporting information

**S1 File. CONSORT 2010 checklist.**
(DOCX)

**S2 File. Raw data.**
(XLSX)

## Acknowledgments

We would like to thank all participants for their contributions. We are especially grateful to the coronary heart disease emergency intervention team and the catheterization staff of the Department of Cardiology of Liaocheng People's Hospital for their selfless dedication in this study.

## Author contributions

**Conceptualization:** Zuolin Fu.

**Data curation:** Hongxu Li, Huawei Dong, Shilin Bao, Xuemei Wang.

**Formal analysis:** Hongxu Li, Shilin Bao, Zuolin Fu.

**Investigation:** Huawei Dong, Shilin Bao.

**Methodology:** Zuolin Fu.

**Project administration:** Zuolin Fu.

**Supervision:** Xuemei Wang.

**Writing – original draft:** Hongxu Li, Xuemei Wang.

**Writing – review & editing:** Zuolin Fu.

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
