## [Decision Letter · Decision Letter 0]

29 Jul 2025

Dear Dr. fu,

We look forward to receiving your revised manuscript.

Kind regards,

Hean Teik Ong, FRCP, FACC

Academic Editor

PLOS ONE

2. In the online submission form, you indicated that your data will be submitted to a repository upon acceptance.  We strongly recommend all authors deposit their data before acceptance, as the process can be lengthy and hold up publication timelines. Please note that, though access restrictions are acceptable now, your entire minimal  dataset will need to be made freely accessible if your manuscript is accepted for publication. This policy applies to all data except where public deposition would breach compliance with the protocol approved by your research ethics board. If you are unable to adhere to our open data policy, please kindly revise your statement to explain your reasoning and we will seek the editor's input on an exemption.

3. We note that you have selected “Clinical Trial” as your article type. PLOS ONE requires that all clinical trials are registered in an appropriate registry (the WHO list of approved registries is at https://www.who.int/clinical-trials-registry-platform/network/primary-registries " https://www.who.int/clinical-trials-registry-platform/network/primary-registries and more information on trial registration is at http://www.icmje.org/about-icmje/faqs/clinical-trials-registration/ ). Please state the name of the registry and the registration number (e.g. ISRCTN or ClinicalTrials.gov ) in the submission data and on the title page of your manuscript. a) Please provide the complete date range for participant recruitment and follow-up in the methods section of your manuscript. b) If you have not yet registered your trial in an appropriate registry, we now require you to do so and will need confirmation of the trial registry number before we can pass your paper to the next stage of review. Please include in the Methods section of your paper your reasons for not registering this study before enrolment of participants started. Please confirm that all related trials are registered by stating: “The authors confirm that all ongoing and related trials for this drug/intervention are registered”. Please see http://journals.plos.org/plosone/s/submission-guidelines#loc-clinical-trials for our policies on clinical trials.

4. Please amend your manuscript to include your abstract after the title page.

5. Please include a caption for figure 1.

Additional Editor Comments:

Please make major revision to the article to address reviewer comments.

Reviewers' comments:

Reviewer's Responses to Questions

**Comments to the Author**

1. Is the manuscript technically sound, and do the data support the conclusions?

Reviewer #1: Partly

Reviewer #2: Yes

2. Has the statistical analysis been performed appropriately and rigorously?

Reviewer #1: Yes

Reviewer #2: Yes

3. Have the authors made all data underlying the findings in their manuscript fully available?

Reviewer #1: Yes

Reviewer #2: Yes

4. Is the manuscript presented in an intelligible fashion and written in standard English?

Reviewer #1: Yes

Reviewer #2: Yes

Reviewer #1: Was a baseline LDL cholesterol level available before treatment of statin and evolocumab was initiated? Even though there is a difference in LDL after treatment, whether it could be accounted for by differences in baseline LDL cholesterol level.

Was other markers of cardiac injury like CK/CKMB measured and were they different between the groups. IL 6 is not easily measured - was CRP measured and if so, was there any difference between groups.

A possible reason for the results to be less significant is because only 1 dose of evolocumab was given. If this was continued, then the LDL lowering effect should still be significant and this could translate into better outcomes for the treatment arm. Any explanation why only 1 dose was given.

Reviewer #2: Interesting paper. Some issues should be added

Abstract: probably for a misprint problem shoild be totally rewritten

methods: primary end point is the end point on which sample size calculatio is performed. Please check

methods/reults: more details about assessment of angina should be stated.

discussion: recently bempedoic acid has shown some potentiality. Please quote on PMID: 38017541 and discuss also the potential itneraction of this treatment.

**Do you want your identity to be public for this peer review?** For information about this choice, including consent withdrawal, please see our Privacy Policy

Reviewer #1: No

Reviewer #2: **Yes: ** Fabrizio D'Ascenzo

---

## [Author Response · Author response to Decision Letter 1]

28 Aug 2025

PONE-D-25-32807

The efficacy of preoperative evolocumab-rosuvastatin combination therapy in patients with ST-elevation myocardial infarction

Dear Editor and Reviewers,

We sincerely appreciate the time and effort dedicated to reviewing our manuscript and providing constructive feedback. We have carefully addressed each comment to improve the quality and clarity of our work. Below, we provide a point-by-point response to the editor reviewers’ comments, with corresponding revisions highlighted in the revised manuscript (tracked changes file).

Response to Editor :

1.deposit laboratory protocols in protocols.io:

It's a good suggestion to deposit laboratory protocols in protocols.we have deposited our laboratory protocols in protocols.io to enhance the reproducibility of results. 

DOI: dx.doi.org/10.17504/protocols.io.n2bvje6mwgk5/v1

(Private link for reviewers:https://www.protocols.io/private/107F70C8702E11F08A690A58A9FEAC02 to be removed before publication.)

2.meets PLOS ONE's style requirements

We have tried to ensure that our manuscripts meet PLOS ONE's style requirements, including document naming requirements and modified accordingly. However, due to our limited ability, please assist in correcting any inadequacies, thank you again for your efforts.

3.data will be submitted to a repository:

Thanks for the reminder, We have now uploaded our complete minimal dataset to Figshare, a PLOS-approved repository:

· Repository: Figshare

· DOI: 10.6084/m9.figshare.29986402

4.Clinical Trial registered and provide the complete date range for participant recruitment and follow-up

We have stated the name of the registry and the registration number in the submission data and on the title page of the manuscript and the reasons for not registering this study before enrolment of participants started in the Methods section. we confirm that all related trials are registered by stating: The authors confirm that all ongoing and related trials for this drug/intervention are registered.

The trial was registered at the Chinese Clinical Trial Registry (ChiCTR2500099498).

This trial was not pre-registered due to initial failure to fully recognize the importance of early registration and lack of familiarity with the registration process. However, a retrospective registration was conducted at the Chinese Clinical Trials Registry (ChiCTR2500099498) before data was analysised, and all data were ensured complete, transparent, and non-selective reporting.The authors confirm that all ongoing and related trials for this drug/intervention are registered.

We have provided the complete date range for participant recruitment and follow-up in the methods section. The dates of start and end of the recruitment period for this study are July 23, 2023 and April 22, 2024, respectively. The last follow-up date was October 26, 2024.

5.amend manuscript to include your abstract after the title page

Thank you for the reminder, we have done as you reminded.

6.include a caption for figure 1

Thank you for your suggestion, we have added a caption for figure 1.

Flow of participants through each stage of the randomized trial. The diagram adheres to the CONSORT 2010 statement.A total of 112 acute ST-elevation myocardial infarction (STEMI) patients were screened between July 2023 and April 2024 at Liaocheng People's Hospital. After exclusions (n=32), 80 patients were randomized to intervention (n=40) or control (n=40). All participants completed the 6-month follow-up and were included in the intention-to-treat�ITT� analysis.

Ethical approval: Institutional Ethics Committee of Liaocheng People's Hospital (LCSY-2023009).

Trial registration: Chinese Clinical Trial Registry (ChiCTR2500099498).

7.the reviewer comments include a recommendation to cite specific previously published works,

We have reviewed and evaluated the publication and determine that it is partly relevant and should be cited. “Recent trials with bempedoic acid (13) have demonstrated additional LDL-C lowering options for statin-intolerant patients. While our study focused on PCSK9 inhibition, the potential synergistic effects of combining bempedoic acid with evolocumab warrants investigation, particularly given their complementary mechanisms targeting cholesterol biosynthesis and clearance respectively. Future studies should explore whether triple therapy (statin+bempedoic acid+PCSK9 inhibitor) could provide incremental benefits in acute coronary syndromes."

Response to Reviewer 1:

1.Baseline LDL Cholesterol Levels:

Thank you for raising this important point. Due to the acute onset of acute myocardial infarction, the patient's diet was uncertain at the time of visit, and the emergency examination items in our hospital did not include LDL cholesterol, and LDL-C testing was not performed before starting statin and elolumab treatment. Considering that the grouping was randomized and there was no significant difference in basic data between the two groups, it is speculated that there may be no significant difference in baseline LDL-C between the two groups, and the difference in LDL-C after treatment is mainly related to treatment. Of course, if there is baseline data, it may be more illustrative, and we will take this into account in future studies.

2. Markers of Cardiac Injury (CK/CKMB) and Inflammation

We agree that additional biomarkers could provide further insights. TNI instead of CK and CK-MB levels was measured during the study, and no significant differences were observed between the groups�CRP was also assessed as a more accessible inflammatory marker, and no significant differences were observed between the groups (data now included in Table 6).

Table6: CRPand TNI Changes on Day 1

control group treatment group z p

CRP(mg/L) 7.52�3.24�20.10� 10.14�3.01�17.69� -0.178 0.859

TNI(ng/ml) 8.00(4.70�14.00) 7.55(3.03�14.25) -0.236 0.814

3. Single-Dose Evolocumab and Treatment Duration:

Thanks for the suggestion. A possible reason for the results to be less significant is because only 1 dose of evolocumab was given. If this was continued, then the LDL lowering effect should still be significant and this could translate into better outcomes for the treatment arm. This has been confirmed by the FORIER study which supports sustained treatment. On the basis of guideline-guided drug therapy, elolumab combined with statin intensive lipid-lowering and anti-inflammatory therapy was applied before emergency PCI to observe the efficacy of a single dose. This is the novelty of this study.

Response to Reviewer 2:

1. Abstract Misprint:

We apologize for the typographical errors in the abstract. We have completely rewritten the abstract to eliminate any potential misprints and improve clarity. The revised version now follows a structured format with clear Background/Methods/Results/Conclusions sections, includes all key numerical findings with precise P-values, and maintains balanced interpretation of results. The clinical trial registration number has been prominently displayed.

2. Primary Endpoint Clarification:

Thank you for this important methodological observation. We confirm that the primary endpoint (6-month MACE) was indeed the basis for our sample size calculation, as stated in the Methods section (2.5 Statistical Analysis). We have now added explicit confirmation: "Sample size was calculated based on the primary endpoint of 6-month MACE incidence, assuming 80% power to detect a 25% absolute reduction between groups (α=0.05, two-sided).

3. Angina Assessment Details:

Additional details on angina assessment (e.g., CCS classification, symptom diaries) have been included in the Methods and Results sections to improve transparency. We have expanded both Methods and Results sections regarding angina assessment:

- Methods (2.4 Outcome Measures): Added "Angina was assessed according to CCS classification criteria through scheduled clinic visits and documented using standardized case report forms."

- Results (3.2.4): Added "All reported angina episodes were physician-adjudicated and required typical symptoms of ischemia with or without objective evidence of ischemia (ECG changes or imaging evidence

4. Bempedoic Acid and Novel Therapies:

We appreciate this valuable suggestion and have substantially expanded the Discussion section to incorporate recent evidence on bempedoic acid:

- Added new paragraph: "Recent trials with bempedoic acid (PMID: 38017541) have demonstrated additional LDL-C lowering options for statin-intolerant patients. While our study focused on PCSK9 inhibition, the potential synergistic effects of combining bempedoic acid with evolocumab warrants investigation, particularly given their complementary mechanisms targeting cholesterol biosynthesis and clearance respectively. Future studies should explore whether triple therapy (statin+bempedoic acid+PCSK9 inhibitor) could provide incremental benefits in acute coronary syndromes."

Additional Revisions

1. Provide rationale for the chosen rosuvastatin dose (10 mg). We added justification in the Introduction (page 3, paragraph 2):

Medium-dose rosuvastatin (10 mg) was selected to balance efficacy and safety in Chinese populations, as higher doses may increase hepatotoxicity risk without proven incremental benefits in acute settings [4].

2. Discuss the clinical relevance of early LDL-C reduction in STEMI. Expanded in Discussion (page 9, paragraph 2):

Rapid LDL-C reduction within 24 hours may attenuate plaque inflammation and microvascular dysfunction, potentially explaining the observed STR improvement.

3.Grammar/Clarity: Polished language throughout (e.g., "postoperative day 1" → "post-PCI day 1").

4. All changes are tracked in the revised manuscript, and new references/citations have been added as appropriate.

We believe these revisions have strengthened the manuscript and addressed all critiques. Thank you for considering our work for publication in PLOS ONE.

Thank you again for your time and expertise.

Sincerely,

Zuo-lin Fu, on behalf of the authors

Email: fzl04@163.com

Attachments:

1. *Revised Manuscript with Track Changes*

2. *Manuscript (clean version)*

3. *Response to Reviewers*

---

## [Decision Letter · Decision Letter 1]

14 Nov 2025

Dear Dr. fu,

We look forward to receiving your revised manuscript.

Kind regards,

Kamal Sharma

Academic Editor

PLOS ONE

Journal Requirements:

Additional Editor Comments:

Hello,

As all the reviewer has accepted the changes made in the manuscript, revised manuscript can be accepted after the minor revision has been addressed of one of the reviewers.

Thanks

Reviewers' comments:

Reviewer's Responses to Questions

**Comments to the Author**

Reviewer #1: All comments have been addressed

Reviewer #2: All comments have been addressed

Reviewer #3: (No Response)

Reviewer #4: All comments have been addressed

2. Is the manuscript technically sound, and do the data support the conclusions?

Reviewer #1: Yes

Reviewer #2: (No Response)

Reviewer #3: Yes

Reviewer #4: Yes

3. Has the statistical analysis been performed appropriately and rigorously?

Reviewer #1: I Don't Know

Reviewer #2: (No Response)

Reviewer #3: Yes

Reviewer #4: Yes

4. Have the authors made all data underlying the findings in their manuscript fully available?

Reviewer #1: Yes

Reviewer #2: (No Response)

Reviewer #3: Yes

Reviewer #4: Yes

5. Is the manuscript presented in an intelligible fashion and written in standard English?

Reviewer #1: Yes

Reviewer #2: (No Response)

Reviewer #3: Yes

Reviewer #4: Yes

Reviewer #1: No additional comments

Reviewer #2: (No Response)

Reviewer #3: (No Response)

Reviewer #4: The authors have presented a well-structured and comprehensive study that have clinical relevance. The findings are relevant for both cardiology and nursing audiences.

Grammatical/Typos:

• Proof read the overall article again, and correct any grammatical error that might be in place.

Narrative:

• Page 1-3, line 25-84: The introduction builds good context on CAD and PCI but lacks specific evidence linking psychological adaptation to clinical outcomes. Add a few sentences citing studies that report the poor resilience leading to worse cardiac rehabilitation adherence or higher anxiety.

• Page 18, 174-178: Conclusion is good but could be more impactful via adding statements like uture longitudinal studies should evaluate how interventions that strengthen coping skills can enhance resilience and reduce cardiovascular event recurrence.

Methods/Results:

• Page 13, Line 66: Clear structure but lacks details on scale validation and sampling. Needs to inform the handling of missing data or incomplete questionnaire.

**Do you want your identity to be public for this peer review?** For information about this choice, including consent withdrawal, please see our Privacy Policy

Reviewer #1: No

Reviewer #2: **Yes: ** Fabrizio D'Ascenzo

Reviewer #3: No

Reviewer #4: No

---

## [Author Response · Author response to Decision Letter 2]

19 Nov 2025

PONE-D-25-32807R1

The efficacy of preoperative evolocumab-rosuvastatin combination therapy in patients with ST-elevation myocardial infarction

Dear Dr. Kamal Sharma and Reviewers,

Thank you for giving us the opportunity to submit a revised version of our manuscript titled "The efficacy of preoperative evolocumab-rosuvastatin combination therapy in patients with ST-elevation myocardial infarction" (Manuscript ID: PONE-D-25-32807R1). We sincerely appreciate the editors and reviewers for your time and valuable comments, which have greatly helped us in improving the quality of our manuscript.

We have carefully addressed the points raised by the reviewers. The changes made in the manuscript are highlighted in the "Revised Manuscript with Track Changes" file. An additional clean version of the revised manuscript has also been submitted.

Response to Editor :

1.deposit laboratory protocols in protocols.io:

It's a good suggestion to deposit laboratory protocols in protocols.we have deposited our laboratory protocols in protocols.io to enhance the reproducibility of results. 

DOI: dx.doi.org/10.17504/protocols.io.n2bvje6mwgk5/v1

(Private link for reviewers:https://www.protocols.io/private/107F70C8702E11F08A690A58A9FEAC02 to be removed before publication.)

2.review reference list to ensure that it is complete and correct

We have reviewed the references in our manuscript. While the reference list is well-structured and as of November 17, 2025, no articles have been marked as “Retracted” or “Expression of Concern” one minor error is identified and corrected in the author list in Reference 10.

Response to Reviewer #4:

We thank the reviewer for their positive feedback and valuable suggestions to further strengthen our manuscript. We have addressed the specific comments as follows:

1. Comment: "Page 1-3, line 25-84: The introduction builds good context on CAD and PCI but lacks specific evidence linking psychological adaptation to clinical outcomes. Add a few sentences citing studies that report the poor resilience leading to worse cardiac rehabilitation adherence or higher anxiety."

· Response: We thank the reviewer for this insightful suggestion. While our study primarily focuses on the pharmacological intervention, we recognize the importance of the patient's psychological state in overall recovery. We have now added a sentence in the Introduction section to acknowledge this aspect and cite relevant literature:

"Beyond pharmacological therapy, a patient's psychological adaptation, such as low resilience, has been linked to poorer adherence to cardiac rehabilitation and higher levels of anxiety, which may indirectly influence long-term clinical outcomes [2]"

[2] Bertolín-Boronat C, Marcos-Garcés V, Merenciano-González H, Martínez Mas ML, Climent Alberola JI, Perez N, López-Bueno L, Esteban Argente MC, Valls Reig M, Arizón Benito A, Payá Rubio A, Ríos-Navarro C, de Dios E, Gavara J, Jiménez-Navarro MF, Chorro FJ, Sanchis J, Bodi V. Depression, Anxiety, and Quality of Life in a Cardiac Rehabilitation Program Without Dedicated Mental Health Resources Post-Myocardial Infarction. J Cardiovasc Dev Dis. 2025 Mar 4;12(3):92. doi: 10.3390/jcdd12030092.

2. Comment: "Page 18, 174-178: Conclusion is good but could be more impactful via adding statements like 'Future longitudinal studies should evaluate how interventions that strengthen coping skills can enhance resilience and reduce cardiovascular event recurrence.'"

· Response: We agree that this is an important direction for future research. We have expanded the Conclusion section to include this perspective:

"Furthermore, future longitudinal studies should explore how interventions combining intensive lipid-lowering therapy with psychosocial support, such as those aimed at strengthening coping skills and resilience, could potentially enhance overall patient outcomes and reduce cardiovascular event recurrence."

3. Comment: "Page 13, Line 66: Clear structure but lacks details on scale validation and sampling. Needs to inform the handling of missing data or incomplete questionnaire."

· Response: We thank the reviewer for pointing this out. We have added details in the Methods section (2.5 Statistical Analysis) regarding data handling:

"The analysis followed the intention-to-treat principle. For the primary and secondary endpoints, there were no missing data as all 80 randomized participants completed the 6-month follow-up. Any incomplete questionnaire items in the case report forms were clarified by direct contact with the attending physician before database locking."

4. Comment: "Proof read the overall article again, and correct any grammatical error that might be in place."

· Response: We have thoroughly proofread the entire manuscript to correct grammatical errors and improve language clarity. This has been done throughout the "Revised Manuscript with Track Changes."

Response to Reviewers #1, #2, and #3:

We thank the reviewers for their previous comments,which have been adequately addressed in the prior revision round, and for their current acceptance of the manuscript.

We believe these revisions have strengthened the manuscript and addressed all critiques. Thank you for considering our work for publication in PLOS ONE.

Thank you again for your time and expertise.

Sincerely,

Zuo-lin Fu, on behalf of the authors

Email: fzl04@163.com

Attachments:

1. *Revised Manuscript with Track Changes*

2. *Manuscript (clean version)*

3. *Response to Reviewers*

---

## [Decision Letter · Decision Letter 2]

9 Dec 2025

The efficacy of preoperative evolocumab-rosuvastatin combination therapy in patients with ST-elevation myocardial infarction

PONE-D-25-32807R2

Dear Dr. fu,

We’re pleased to inform you that your manuscript has been judged scientifically suitable for publication and will be formally accepted for publication once it meets all outstanding technical requirements.

Kind regards,

Kamal Sharma

Academic Editor

PLOS ONE

Additional Editor Comments (optional):

Reviewers' comments:

Reviewer's Responses to Questions

**Comments to the Author**

Reviewer #2: All comments have been addressed

2. Is the manuscript technically sound, and do the data support the conclusions?

Reviewer #2: Yes

3. Has the statistical analysis been performed appropriately and rigorously?

Reviewer #2: Yes

4. Have the authors made all data underlying the findings in their manuscript fully available?

Reviewer #2: Yes

5. Is the manuscript presented in an intelligible fashion and written in standard English?

Reviewer #2: Yes

Reviewer #2: all comments have been addressed and authors should be complimented for performing such an interesting study

**Do you want your identity to be public for this peer review?** For information about this choice, including consent withdrawal, please see our Privacy Policy

Reviewer #2: **Yes: ** Fabrizio D'Ascenzo

---

## [Editor Report · Acceptance letter]

PONE-D-25-32807R2

PLOS One

Dear Dr. fu,

I'm pleased to inform you that your manuscript has been deemed suitable for publication in PLOS One. Congratulations! Your manuscript is now being handed over to our production team.

Kind regards,

on behalf of

Dr. Kamal Sharma

Academic Editor

PLOS One